# Racial discrimination, self-efficacy, and oral health behaviours in adolescents

**Sanaz Bohlouli[1], Samin Dolatabadi[2], Babak Bohlouli[3], Maryam Amin [3] ***

**1** Department of Biological Sciences, University of Alberta, Edmonton, Canada, **2** School of Medicine, University of Calgary, Calgary, Canada, **3** School of Dentistry, University of Alberta, Edmonton, Canada

☯ These authors contributed equally to this work.
* maryam.amin@ualberta.ca

**Data Availability Statement:** Data are available from the University of Alberta Institutional Data Access / Ethics Committee (contact Research Ethics Office via reoffice@ualberta.ca) for

## Abstract

To examine the mediation effect of discrimination on the association of self-efficacy and oral health behaviours among adolescents. A cross sectional study of adolescents aged 12 to 18 years who were recruited from the University outpatient dental clinic were asked to complete a questionnaire consisting of: demographics (12 items), oral health behaviours (7 items), general self-efficacy (10 items) and self-efficacy for self-care (SESS, 15 items). Perceived discrimination was assessed if the adolescent had ever been treated unfairly based on their race. Perceived discrimination was assessed if the adolescent had ever been treated unfairly based on their race. Using pathway analyses, the relationship between oral health behaviours, self-efficacy, and discrimination was explored. Mediation and hierarchal logistic regression analyses were conducted. Of 252 participants, mean (SD) age was 14 (1.8) years old. 60% were female, 81% were born in Canada, 56% identified themselves as White, and 20% perceived discrimination. Mean score of all task-specific self-efficacies were significantly different within respective oral health behaviour categories (P-value <0.001). Of demographics, age and ethnicity (White) were significantly associated with discrimination (OR = 1.25: 95% CI; 1.06–1.48 and OR = 0.29: 95% CI; 0.15–0.55, respectively). Perceived discrimination was positively associated with higher sugar consumption and mediate the association between diet self-efficacy and adolescent's dietary behaviour. Significant mediation effect of perceived discrimination on the association of diet specific self-efficacy and diet oral health behaviour was observed. Oral health behaviours were self-reported which may have influenced the results.

## Introduction

Dental health is the most prevalent unmet need among adolescents [1]. When self-care practices switch from parents to adolescents, dental utilization rates begin to drop [2]. This leads to worse oral health outcomes. Health Canada [3] reports that the average Decayed, Missing, Filled Teeth (DMFT) score for adolescents is 2.49 and the National Center for Health Statistics [4] reports that three in five adolescents have experienced dental caries in their permanent teeth. According to Baju et al, oral health is defined as the well-being and ability to perform functions such as eat, talk, and smile [5]. In healthcare, racial discrimination is referred to an individual's perception of unfair delivery of care in a medical setting because of race, color, or nationality [6, 7]. Discrimination can occur at different levels including individual,

researchers who meet the criteria for access to confidential data. The data includes sensitive information such as date of birth, therefore, it needs parents or guardians consent to share.

**Funding:** The authors received no specific funding for this work.

**Competing interests:** The authors have declared that no competing interests exist.

institutional, and structural level [8]. Discrimination can lead to poor general and oral health, increased chronic health problems such as hypertension and cardiovascular disease, higher vulnerability towards stroke, heart attack, and diabetes, higher levels of depression and anxiety, and risky behaviours (such as drinking and smoking) [9–14].

Racial disparities have been reported in utilization of dental services. In addition to socio-economic and cultural aspects, ethnicity itself was found to be strongly associated with dental care usage among adolescents in New Zealand [15], which could be a result of racial discrimination. The emotional impact reported as an important predictor of use of dental services among American adults in a recent study [16]. Nonetheless, reports on the impact of racial discrimination on oral health are contradictory.

In healthcare, self-efficacy refers to a patient's perception regarding whether they can take the necessary actions to improve their health status [17]. In the oral health field, several studies have also investigated the relationship between task-specific or general self-efficacy and oral health behaviour and outcomes. In a sample of college students, for example, McCaul et al found that stronger toothbrushing and flossing self-efficacy beliefs were significantly related to frequency of these activities [18]. In another study, it was also found that toothbrushing and flossing self-efficacy scores were significantly associated with frequency of dental visits and dental knowledge [19].

According to the social cognitive theory, developed by Albert Bandura, individual's behaviour is influenced by core determinants, which includes self-efficacy and environmental factors such as discrimination [20]. According to Hansen, discrimination is linked to various health impairments such as chronic muscle pain, diabetes, and metabolic syndrome [21]. Previous research has also determined the role of discrimination as a psychosocial stressor that contributes to racial inequalities in oral health outcomes. Furthermore, it highlights the importance to go beyond the individual-level factors like socioeconomic status and genetic makeup and to consider wider sociohistorical processes such as systematic racism, economic exploitation, social stigmatization, and political marginalization in racial health outcomes [22].

Similarly, the study conducted by Schuch et al with Australian adults demonstrated that discrimination negatively affected oral health outcomes; however, the impacts were more pronounced in socially marginalized groups [23]. Furthermore, after controlling for other risk factors such as income, education, age, sex, Marital status, employment status, and type of social insurance, Hispanic individuals with perceived discrimination were less likely to visit a dentist and more likely to have tooth loss compared to non-Hispanic white individuals [24].

Another study conducted by Christinia et al also indicated the importance of addressing racism and its physiological impacts on oral health in dental education in order to promote oral health equity, regardless of ethnic backgrounds [25]. Yet, the mediation effect of discrimination has not been fully investigated in oral health. Therefore, the aims of this study were to examine the mediation effect of discrimination on the association of self-efficacy and oral health behaviours among adolescents.

## Methods

### Study settings and participants

In this cross-sectional study, participants were recruited from the University of Alberta dental clinic in 2021. Aged 12-18-year-old who could understand English and had no significant physical or mental disabilities were included in the study. Affordable costs and high volumes of visits were the reasons to choose the University dental clinic to recruit participants. The trained research team members explained the purpose of the study to the participants and their families while they were waiting for their appointment. Signed consent and assent forms

were obtained in advance from parents or guardians to collect data from participants. Those variables with more than 10% missing were not included in multivariant analysis. Anonymous code numbers are used to identify participants and only the researchers had access to them.

The study protocol was approved by the ethics board of the University of Alberta (Ethics approval number was pro00077682).

## Data collection and procedure

Given the average yearly number of visits at the university dental clinic, with 95% confidence interval and 5% margin of error desired sample size was 278, however 252 patients agreed to complete the questionnaires. Which is close to the ideal number.

A questionnaire consisting of three sections were completed by adolescents (S1 Table). In part one, demographic data was collected for children and their families. The second section of questionnaire was about four oral health behaviours: frequency of consuming sugary food or drink; frequency of toothbrushing; time of last dental visit and reason (regular check-up, non-urgent or urgent dental problems). In addition to oral health behaviours, adolescents self-rated oral health status was evaluated based on one question: "How do you think your oral health is?" with the possible options being "very good," "good," "fair," and "poor."

General and task-specific self-efficacy were investigated in section three using a validated version of general self-efficacy (GSE) scale, a questionnaire that was developed originally in German but has been adapted to 28 languages including English [26]. The scale is composed of 10 questions on a 5-point Likert scale. The final score is a sum of all items ranging from 10–50. Self-efficacy for self-care scale (SESS) was used to measure task-specific self-efficacy [27]. This validated scale contains 15 questions with 3 subscales assessing self-efficacy for tooth-brushing, dietary habits, and dental visits on a 5-point Likert scale similar to GSE scale. Perceived discrimination was measured by: "Have you ever been treated unfairly or discriminated against based on your race?".

Using AMOS, a visual program for path analysis, the relationships between self-efficacy, discrimination, and oral health behaviors were investigated to find whether discrimination had any mediation effect on the association between self-efficacy and oral health behaviors or self-efficacy mediated the other two components association. The power of study based on type 1 error was reasonable based on sample size.

## Data analysis

Percentages represented categorial variables, and continuous variables were identified by means, standard deviations, and ranges. Univariate analyses were performed to examine the impact of any demographic factors on the outcomes of interest. Mediation and hierarchal logistic regression analyses were conducted to find if discrimination had any mediation effect on associations between task-specific self-efficacy and general self-efficacy and the respective oral health behaviour. Internal consistency coefficient of self-efficacy items was 0.90. Cronbach's alpha was also greater than 0.80 suggesting that the scales were homogenous. Statistical Package for Social Sciences (SPSS for Windows, version 24.0; SPSS Inc., Chicago, Ill., USA) program and statistical significance levels was defined using 95% confidence interval p-values less than 0.05.

## Variables of interest

Outcome variables included oral health's behaviour. In this study, discrimination and self-efficacy score were predictors of the model and the confounding factors included age, gender, location of birth, ethnicity, and income.

## Results

A total of 252 participants were included in the analyses with mean (SD) age of 14 (1.8) years old. We used a convenience sampling approach and almost all adolescents approached completed the survey. Therefore, the participants' rate was about 90%. Sixty percent of participants were girl and age were not statistically different between males and females (p-value > 0.05). Eighty-one percent of participants were born in Canada and 56% were self-identified as White (Caucasian). Of those participants who responded, 65% of mothers had college or university education and 61% had dental insurance (insurance variable has 55% missing data). Participants' demographics are presented in **Table 1**. About 30% of participants reported their oral health as not good and about 20% reported some experience of discrimination. Significant correlations were found among general and all task-specific self-efficacies (**Table 2**). Of all the participants, 52 individuals 61% female) with mean (SD) age of 15 (2) years reported discrimination. This group was on average one year older than those who did not report discrimination (P-value <0.05). No statistically significant difference was found for sex and insurance coverage between the two groups (P-value>0.05).

### General and task specific self-efficacy and respective behaviour

General and task-specific self-efficacy mean scores difference between categories of oral health behaviours were presented in **Table 3**. Mean score of all task-specific self-efficacies were significantly different within respective oral health behaviour categories (P-value <0.001). Mean

**Table 1. Participant characteristics (N = 252).**

| Characteristics | N (%) |
|---|---|
| No of children in family | |
| 1 | 18 (7) |
| 2 | 100 (40) |
| ≥ 3 | 134 (53) |
| Mother's level of education | |
| High school or lower | 63 (25) |
| College/University | 165 (65) |
| Monthly income level | |
| Less than $1,999 | 7 (3) |
| $2,000 - $3,999 | 38 (15) |
| More than $4,000 | 85 (34) |
| Child gender | |
| Female | 152 (60) |
| Child age (mean, SD, range) | 14 (1.77) (12–18) |
| Child birth place | |
| Out of Canada | 49(19) |
| Living with | |
| Single parent | 60 (24) |
| Both parents | 192 (76) |
| Ethnicity | |
| Others | 110 (44) |
| Whites | 142 (56) |
| Child dental insurance | |
| No insurance | 56 (22) |
| Has insurance | 154 (61) |

**Table 2. Task specific and general self-efficacy correlation.**

|  | S.E. brushing | S.E. diet | S.E. visit | S.E. general |
|---|---|---|---|---|
| S.E. brushing | 1 |  |  |  |
| S.E. diet | 0.44* | 1 |  |  |
| S.E. visit | 0.43* | 0.33* | 1 |  |
| S.E. general | 0.54* | 0.40* | 0.35* | 1 |

*significant (P-value<0.05)

score of general self-efficacy was significant only in frequency of toothbrushing behaviour. General and all task-specific self-efficacies showed different mean score within self-rated oral health categories. In the adjusted analysis for the participants' demographics, odds ratio (OR) of brushing more than twice per day increased by any unit increase in toothbrushing self-efficacy (OR = 1.30; 95% CI: 1.13–1.51). For the diet, with any unit increase in diet self-efficacy, the risk of consuming sugar decreased 9% (OR = 0.89; 95% CI: 0.83–0.95). In addition, participants with higher dental visit self-efficacy had 11% higher chance of a dental checkup (OR = 1.11; 95% CI; 1.03–1.20). When examining general self-efficacy with oral health behaviour, only toothbrushing frequency was significantly associated with GSE (OR = 1.05: 95%CI; 1.01–1.09).

## Perceived discrimination

Mediation between perceived discrimination, general and task-specific self-efficacy, and oral health behaviours were tested using IBM® SPSS® Amos. CMIN/DF for the default model was 7.2, providing close to reasonable goodness of fit (CMIN = Chi-square value)

**Table 3. Self-efficacy means score difference by oral health behaviour.**

| Outcomes | N (%) | Self-efficacy | | | | | |
|---|---|---|---|---|---|---|---|
|  |  | Tooth brushing | Dietary habits | Dental visits | P-value | General | P-value |
| Frequency of dental brushing |  |  |  |  |  |  |  |
| < 2x/day | 77 (30) | 17.8 | NA |  | 0.001 | 36.1 | 0.001 |
| ≥ 2x/day | 175 (70) | 20.6 |  |  |  | 3.7 |  |
| Sugar-intake frequency |  |  |  |  |  |  |  |
| < 1x/day | 130 (52) | NA | 16.8 | NA | 0.001 | 38.6 | > 0.05 |
| ≥ 1x/day | 122 48) |  | 15.2 |  |  | 37.3 |  |
| Utilization of dental services (last year) |  |  |  |  |  |  |  |
| Yes | 215(85) | NA |  | 19.6 | 0.001 | 38.3 | > 0.05 |
| No | 37 (15) |  |  | 17.7 |  | 35.9 |  |
| Pattern of dental attendance |  |  |  |  |  |  |  |
| Dental problem | 17 (6.7) | NA | 17.9 |  | 0.001 | 37 | > 0.05 |
| Regular check up | 198 (78.5) |  | 19.7 |  |  | 38 |  |
| Self-rated dental health |  |  |  |  |  |  |  |
| Fair or poor | 73 (29) | 17.5* | 14.8* | 17.6* | * sig | 36 | 0.001 |
| Good | 179 (71) | 20.6 | 15.5 | 20 |  | 39 |  |
| Discrimination |  |  |  |  |  |  |  |
| Yes | 52 (20) | 18.4* | 16.3* | 18.5 | * sig | 36.7 | > 0.05 |
| No | 200 (80) | 20 | 14.5 | 19.5 |  | 38.2 |  |

*Statistically significant p-value <0.05, NA: Not applicable

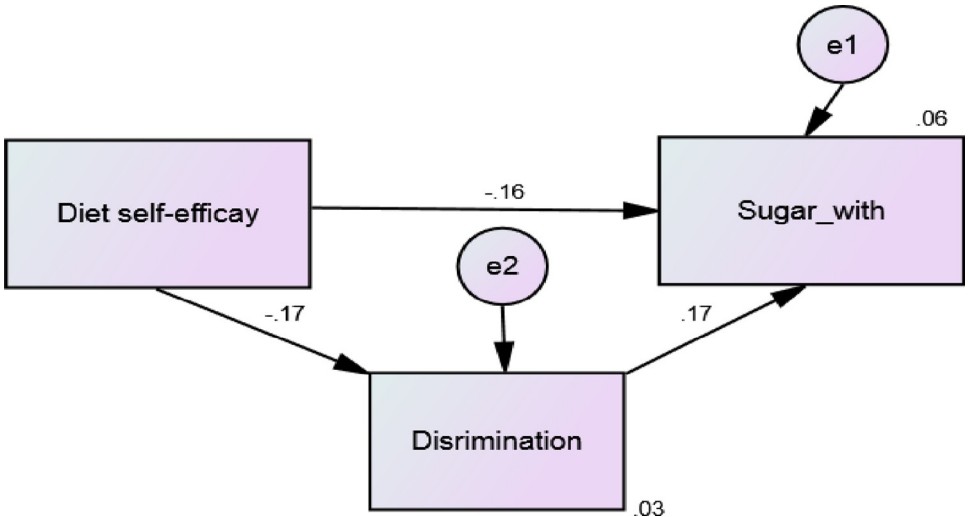

**Fig 1. Mediation relationship between diet specific self-efficacy, discrimination, and diet oral health behaviours.**

(DF = Degree of Freedom). Significant mediation effect of perceived discrimination on the relationship of diet self-efficacy and respective oral health behaviour was observed (P-value = 0.001). While the beta coefficient of direct association of diet self-efficacy and respective oral health behaviour was -0.16, total beta coefficient (direct and in-direct) was– 0.19. Indirect effect was statistically significant (**Fig 1**). In the univariate analyses, of all participants' demographic characteristics, age and ethnicity had significant associations with discrimination (OR = 1.25: 95% CI; 1.06–1.48 and OR = 0.29: 95% CI; 0.15–0.55, respectively) (**Table 4**). Findings illustrated that in those participants experiencing discrimination, risk of sugar consumption was 2.7 times higher (OR = 2.67; 95% CI; 1.40–5.08) than in those who did not report racial discrimination. To explore the mediation effect of discrimination on the association of general and task specific self-efficacies with respective self-efficacies, we found that perceived discrimination mediated the association between diet self-efficacy and sugar consumption. In adjusted hierarchal logistic regression analyses, chi-square for Hosmer and Lemeshow test for the final model was 3.58 and statistically non-significant indicating that model fit the data reasonably well. Perceived discrimination significantly increased the risk of sugar consumption by 2% (OR = 0.89; 95% CI: 0.83–0.95 vs OR = 0.91; 95% CI: 0.84–0.96) (**Table 5**).

**Table 4. Univariate analyses of discrimination with demographics.**

| Discrimination | Demographics | Odds Ratio | 95% CI |
|---|---|---|---|
| | Age | 1.25* | 1.06–1.48 |
| | Male | 0.93 | 0.50–1.75 |
| | Born in Canada | 1.19 | 0.53–2.65 |
| | Ethnicity: white | 0.29* | 0.15–0.55 |
| | Living with both parents | 1.05 | 0.51–2.16 |
| | No of children | 1.06 | 0.65–1.74 |
| | Mother university education | 0.65 | 0.32–1.30 |
| | Income | 0.78 | 0.40–1.53 |
| | Have dental coverage | 1.36 | 0.60–3.08 |

*significant (P-value<0.05)

**Table 5. Results of hierarchal logistic regression analyses.**

| Model (1) (significant demographics were included in the model) | | | | |
|---|---|---|---|---|
| Sugar consumption | | | | |
| | Odds Ratio | Significance | Lower 95% (CI) | Upper 95%(CI) |
| Diet specific self-efficacy | 0.89 | 0.001 | 0.83 | 0.95 |
| Age | 0.81 | 0.001 | 0.69 | 0.94 |
| Self- rated oral health | 1.27 | Non sig | 0.72 | 2.27 |
| Ethnicity | 1.20 | Non sig | 0.71 | 2.02 |
| Model (2) (perceived discrimination was added to the mode) | | | | |
| Sugar consumption | | | | |
| Diet specific self-efficacy | 0.91 | 0.001 | 0.84 | 0.96 |
| Age | 0.77 | 0.001 | 0.65 | 0.90 |
| Self- rated oral health | 1.20 | Non sig | 0.67 | 2.17 |
| Ethnicity | 0.97 | Non sig | 0.56 | 1.67 |
| No Perceived discrimination | 0.34 | 0.001 | 0.16 | 0.70 |

Goodness of fit: Hosmer and Lemeshow test, Chi-square = 3.58, df = 8, P-value = 0.88

## Discussion

Although many studies have provided compelling evidence for the unfair effects that perceived discrimination can have on certain health outcomes [21, 28], limited studies investigated this relation in the oral health domain [29, 30]. In addition, to the best of our knowledge, there is no study to examine conditions that may have mediation effect on the relationship of oral health behaviours and individuals' self-efficacy, known as a predictor of oral health behaviour. In the present study, we examined the association between self-efficacy and oral health behaviours in adolescents. While a significant association was found between toothbrushing, dietary habits, and dental visits self-efficacy (subscales of task-specific self-efficacy) and their respective outcomes (frequency of toothbrushing, sugar-intake, and regular dentist visits), general self-efficacy was significantly associated with the frequency of toothbrushing and participant's self-rated oral health. We also explored the mediation effect of perceived racial discrimination on this relationship. From the three oral health behaviours, perceived discrimination only altered the dietary habits of the adolescents and mediated the association between diet self-efficacy and sugar consumption.

Based on our results, task-specific and general self-efficacy were related to various oral health behaviours in adolescents. For instance, individuals with a higher toothbrushing self-efficacy brushed their teeth more frequently. Also, Anagnostopoulos et al found that self-efficacy beliefs are a strong predictor of tooth brushing frequency, as a result, participants with stronger self-efficacy beliefs brushed their teeth more frequently and had better oral health status [31]. As a result, interventions focusing on enhancing self-efficacy resulted in better behavioural (timing, duration, method) outcomes in participants [32]. In our study, self-efficacy for dietary habits and dental visits influenced their respective outcome. Similarly, a previous study found that, low levels of self-efficacy contribute to infrequent dental visits [33].

No significant association was found between dietary self-efficacy and consumptions of sugar when demographic factors were considered. According to the findings of Jamieson et al, there was an association between self-efficacy and self-rated oral health behaviours among pregnant Aboriginal Australian women after adjusting for sociodemographic, psychosocial, and behavioural factors [34]. This is consistent with our findings except the study was conducted on adult population instead of adolescents. This highlights the significance of self-efficacy in determining health outcomes.

In our study, general self-efficacy was only associated with tooth brushing and not with sugar intake and dental visit. Similarly, general self-efficacy did not predict completion of periodontal treatment and dental consultation in another study when they compared it to the task-specific self-efficacy [35]. Wolfe et al also found that general self-efficacy was not associated with plaque index using Dental Coping Beliefs Scale [36]. Based on previous findings, one can conclude general self-efficacy may not be an accurate measure of performance. Contrary to previous studies, we found that general self-efficacy was highly associated with participants' self-rated oral health. Self-rated oral health is both clinically associated with dmft and is socially associated with patient-dentist communication and oral health literacy [37, 38]. Self-rated oral health is a valid measure of people's oral health and is the key factor for better quality of life [37, 39]. In addition, in our study, age and white ethnicity independently had significant associations with perceived discrimination. Similar to a previous study by Deitch et al, the participants who identified themselves as Whites reported less racial discrimination experience compared to the non-white participants [40]. Ethnic identity has been found to act as a protective mechanism to mitigate the effects of racial discrimination on individuals' psychological well-being. Therefore, African American adolescents with weak ethnic identities strongly correlated with higher rates of perception of discrimination leading to young adult drug use [41]. Additionally, age was accounted to be a strong demographic predictor of perceived discrimination in our study; older adolescents reported higher experience of racial discrimination. Similarly, Fisher et al found that 12th graders self-identified as African American reported higher institutional discriminatory distress than 9th-11th graders did [27]. Perhaps, older adolescents become more aware of various racial groups resulting in higher perception of discrimination. This is consistent with findings of Seaton et al who also reported that perceived discrimination becomes clear through development of cognitive strategies such as reasoning and abstract thinking in older adolescents than younger adolescents [42].

In our study, sugar-intake and toothbrushing behaviours were significantly different between the group who reported perceived discrimination and those who did no. Perceived discrimination increased sugar-intake in adolescents [43]. A previous study has found that perceived discrimination promoted unhealthy eating habits in adolescents in order to cope with stress caused by discrimination [44]. Adolescence is considered the most vulnerable phase in life, and the presence of stress resulting from prejudicial treatment makes adolescents more susceptible to engage in unhealthy behaviours as a way to buffer the stress. Adolescents with perceived discrimination, in our study, did not brush their teeth as often compared to those who did not have such a perception. Similarly, Ben et al found that high levels of discrimination reported by pregnant aboriginal Australian women was associated with non-optimal toothbrushing behaviours [45]. This suggests that influence of discrimination is carried on beyond emotional impact.

Our study had some limitations that need to be acknowledged. First, there is some selection bias with the participants in the study. All participants were recruited from the University clinic that reduces the ability to extrapolate our findings to the general population. Second, participants seeking dental care have higher motivation or better access to oral health resources that may distort true associations. Moreover, assessments of oral health behaviour were self-reported, which may have resulted in overestimation or underestimation. Furthermore, it is also important to consider the socioeconomic status of individuals who experience perceived discrimination on their oral health behaviour. In future studies, some monitoring tools such as toothbrushing and dietary charts can be used to accurately record toothbrushing and sugar-intake frequencies rather than patient report that may induce recall bias in the study. Self-rated oral health measure was used as a proxy for oral health status as the data set had no information regarding oral clinical outcomes. Dental caries, periodontal conditions

and other valuable variables should also be considered as an outcome variable. The results of this study are not generalizable to the whole adolescent population because we recruited the participants from dental clinics, therefore, could have a more favourable attitude towards oral health and dental visits and less barrier to access dental care.

## Conclusions

General and oral health-related self-efficacy were associated with oral health-related behaviours such as toothbrushing and sugar-intake among adolescents. Perceived discrimination could significantly medicate association of diet subclass of self-efficacy with its respective behaviour. Thus, assessment of self-efficacy is beneficial and useful in dental practice to identify adolescents with low self-efficacy.

## Supporting information

**S1 Checklist. STROBE checklist.**
(DOCX)

**S1 Table. Research questionnaire.**
(DOCX)

## Acknowledgments

The authors would like to thank the patients and their families who agreed to participate in this study and the dental team for their help with the recruitment.

## Author Contributions

**Conceptualization:** Maryam Amin.

**Data curation:** Sanaz Bohlouli.

**Formal analysis:** Babak Bohlouli.

**Investigation:** Maryam Amin.

**Methodology:** Samin Dolatabadi.

**Project administration:** Maryam Amin.

**Supervision:** Maryam Amin.

**Visualization:** Sanaz Bohlouli, Babak Bohlouli.

**Writing – original draft:** Sanaz Bohlouli.

**Writing – review & editing:** Sanaz Bohlouli, Babak Bohlouli, Maryam Amin.

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
