## [Decision Letter · Decision Letter 0]

28 Apr 2023

PONE-D-23-08590Racial discrimination, self-efficacy, and oral health behaviours in adolescentsPLOS ONE

Dear Dr. Amin,

Thank you for submitting your manuscript to PLOS ONE. After careful consideration, we feel that it has merit but does not fully meet PLOS ONE’s publication criteria as it currently stands. Therefore, we invite you to submit a revised version of the manuscript that addresses the points raised during the review process.

We look forward to receiving your revised manuscript.

Kind regards,

Hadi Ghasemi

Academic Editor

PLOS ONE

Journal Requirements:

Reviewers' comments:

Reviewer's Responses to Questions

**Comments to the Author**

1. Is the manuscript technically sound, and do the data support the conclusions?

Reviewer #1: Partly

Reviewer #2: Yes

Reviewer #3: Yes

2. Has the statistical analysis been performed appropriately and rigorously? 

Reviewer #1: Yes

Reviewer #2: Yes

Reviewer #3: Yes

3. Have the authors made all data underlying the findings in their manuscript fully available?

Reviewer #1: Yes

Reviewer #2: Yes

Reviewer #3: No

4. Is the manuscript presented in an intelligible fashion and written in standard English?

Reviewer #1: Yes

Reviewer #2: Yes

Reviewer #3: Yes

5. Review Comments to the Author

Reviewer #1: Manuscript number: PONE-D-23-08590

Title: Racial discrimination, self-efficacy, and oral health behaviours in adolescents

General comments:

The authors demonstrated that perceived racial discrimination is related to the deterioration of diet self-efficacy and an increase in the intake of sugar-containing food. Potentially, the topic of the manuscript is interesting; however, there are some issues that should be addressed in the manuscript.

Abstract

Please describe the study setting.

Please explain regarding path analysis.

Introduction

P10L87-88: Authors argued that the relationship with mediation effect of discrimination has not yet been fully investigated in oral health. I agree your argument. However, many studies on the relationship between oral health and discrimination were reported, for instance, Racial Inequalities in Oral Health. (J Dent Res. 2018 Jul;97(8):878-886), Intersectionality, racial discrimination and oral health in Australia. (Community Dent Oral Epidemiol. 2021 Feb;49(1):87-94), Perceived racial discrimination partially mediates racial-ethnic disparities in dental utilization and oral health. (J Public Health Dent. 2022 Mar;82 Suppl 1(Suppl 1):63-72, Racial and oral health equity in dental school curricula. (J Public Health Dent. 2022 Mar;82 Suppl 1(Suppl 1):114-122).

Authors should cite the previous reports regarding the relationship, and the author should clearly explain in their research why they conducted the study by stating what has already been discovered and what remains unknown based on previous research.

Methods

Please explain regarding path analysis using Amos.

Results

Please describe the participation rate if you can.

Please provide the Cronbach’s alpha for each scale.

Table 1 should describe the characteristics of each group of participants divided based on whether they have experienced discrimination or not, with statics analysis.

Table 3 Please explain mean of the symbol in footnote.

Figure 1. Did diet self-efficacy have negative influence on discrimination? Please recheck the model for path analysis, and perform goodness-of-fit test for the model.　

Table 5. Please provide data on the goodness of fit of each model in regression analysis.

Discussion

L253-257. I agree with the discussion. However, the authors should also consider the impact of the socioeconomic status of individuals who experience perceived discrimination on their oral health behavior.

L277-278. “Thus, assessment of self-efficacy is beneficial and useful in dental practice to identify those with low self-efficacy.” What does the pronoun “those” indicate? Persons with perceived racial discrimination?

L278-281. “Behavioral interventions…”. The present study has no data for the behavioral interventions. Thus, the statement may be excessive discussion.

Reviewer #2: Thank you for your valuable article

1. Please attach the questionnaire

2. Indicate the study's design in the abstract

3. Provide a specific definition of dental health/oral health

4. You have considered only the frequency of toothbrushing in your study, so how about timing, duration, and method? Why were they not evaluated in the questionnaire?

5. Please describe analytical methods taking account of the sampling strategy

6. in the discussion it's better to give more interpretation of results and discuss the generalizability of your results

Reviewer #3: Overall, this is a clearly written manuscript and the subject of the manuscript is important in dental literature.

Just few points I would like the authors to explain or revise:

1. The sample size calculation method should be indicated in details.

2. The sampling technique used to recruit participants needs to be more explained: how was the sampling performed?

No additional comments for the authors about dual publication, research ethics, or publication ethics.

6. PLOS authors have the option to publish the peer review history of their article (what does this mean?). If published, this will include your full peer review and any attached files.

Reviewer #1: No

Reviewer #2: No

Reviewer #3: No

---

## [Author Response · Author response to Decision Letter 0]

5 Jul 2023

We have addressed all the comments and revised manuscript accordingly.

---

## [Decision Letter · Decision Letter 1]

26 Jul 2023

Racial discrimination, self-efficacy, and oral health behaviours in adolescents

PONE-D-23-08590R1

Dear Dr. Maryam Amin,

We’re pleased to inform you that your manuscript has been judged scientifically suitable for publication and will be formally accepted for publication once it meets all outstanding technical requirements.

Kind regards,

Hadi Ghasemi

Academic Editor

PLOS ONE

Additional Editor Comments (optional):

Reviewers' comments:

Reviewer's Responses to Questions

**Comments to the Author**

1. If the authors have adequately addressed your comments raised in a previous round of review and you feel that this manuscript is now acceptable for publication, you may indicate that here to bypass the “Comments to the Author” section, enter your conflict of interest statement in the “Confidential to Editor” section, and submit your "Accept" recommendation.

Reviewer #1: All comments have been addressed

Reviewer #2: All comments have been addressed

Reviewer #3: All comments have been addressed

2. Is the manuscript technically sound, and do the data support the conclusions?

Reviewer #1: Yes

Reviewer #2: Yes

Reviewer #3: Yes

3. Has the statistical analysis been performed appropriately and rigorously? 

Reviewer #1: Yes

Reviewer #2: Yes

Reviewer #3: Yes

4. Have the authors made all data underlying the findings in their manuscript fully available?

Reviewer #1: No

Reviewer #2: Yes

Reviewer #3: No

5. Is the manuscript presented in an intelligible fashion and written in standard English?

Reviewer #1: Yes

Reviewer #2: Yes

Reviewer #3: Yes

6. Review Comments to the Author

Reviewer #1: Authors addressed all issues which the reviewer #1 have indicated. There are no additional comments for the manuscript.

Reviewer #2: Thank you for your answers,

About this one:

"You have considered only the frequency of tooth brushing in your study, so how about timing,

duration, and method? Why were they not evaluated in the questionnaire?

- Response: Thank you for the comment. While all aspects of tooth brushing practices are

important, we only included the frequency, which is the most common variable used in

oral health studies, to avoid making the questionnaire too long. "

Adding one or two questions to the questionnaire will not make it longer, instead, it will greatly increase the quality of the study.

Reviewer #3: The authors revised their manuscript properly according to the reviewers comments. All comments have been assressed.

7. PLOS authors have the option to publish the peer review history of their article (what does this mean?). If published, this will include your full peer review and any attached files.

Reviewer #1: No

Reviewer #2: No

Reviewer #3: No

---

## [Editor Report · Acceptance letter]

3 Aug 2023

PONE-D-23-08590R1 

Racial discrimination, self-efficacy, and oral health behaviours in adolescents 

Dear Dr. Amin:

I'm pleased to inform you that your manuscript has been deemed suitable for publication in PLOS ONE. Congratulations! Your manuscript is now with our production department. 

Kind regards, 

on behalf of

Dr. Hadi Ghasemi 

Academic Editor

PLOS ONE